# Accuracy Assessment of Percutaneous Pedicle Screw Placement Using Cone Beam Computed Tomography with Metal Artifact Reduction

**DOI:** 10.3390/s22124615

**Published:** 2022-06-18

**Authors:** Yann Philippe Charles, Rawan Al Ansari, Arnaud Collinet, Pierre De Marini, Jean Schwartz, Rami Nachabe, Dirk Schäfer, Bernhard Brendel, Afshin Gangi, Roberto Luigi Cazzato

**Affiliations:** 1Department of Spine Surgery, University Hospital of Strasbourg, 67200 Strasbourg, France; rawan.al-ansari@chru-strasbourg.fr (R.A.A.); arnaud.collinet@chru-strasbourg.fr (A.C.); 2Department of Interventional Radiology, University Hospital of Strasbourg, 67000 Strasbourg, France; pierre.demarini@chru-strasbourg.fr (P.D.M.); jean.schwartz@chru-strasbourg.fr (J.S.); gangi@unistra.fr (A.G.); robertoluigi.cazzato@chru-strasbourg.fr (R.L.C.); 3Department of Image Guided Therapy Systems, Philips Healthcare, 5684 PC Best, The Netherlands; anindita.chatterjea@philips.com; 4Department of Image Formation and Medical Image Acquisition, Philips Research, 22335 Hamburg, Germany; dirk.schaefer@philips.com (D.S.); bernhard.brendel@philips.com (B.B.)

**Keywords:** augmented reality, surgical navigation, cone beam computed tomography, metal artifact reduction algorithm, screw accuracy, image quality

## Abstract

Metal artifact reduction (MAR) algorithms are used with cone beam computed tomography (CBCT) during augmented reality surgical navigation for minimally invasive pedicle screw instrumentation. The aim of this study was to assess intra- and inter-observer reliability of pedicle screw placement and to compare the perception of baseline image quality (NoMAR) with optimized image quality (MAR). CBCT images of 24 patients operated on for degenerative spondylolisthesis using minimally invasive lumbar fusion were analyzed retrospectively. Images were treated using NoMAR and MAR by an engineer, thus creating 48 randomized files, which were then independently analyzed by 3 spine surgeons and 3 radiologists. The Gertzbein and Robins classification was used for screw accuracy rating, and an image quality scale rated the clarity of pedicle screw and bony landmark depiction. Intra-class correlation coefficients (ICC) were calculated. NoMAR and MAR led to similarly good intra-observer (ICC > 0.6) and excellent inter-observer (ICC > 0.8) assessment reliability of pedicle screw placement accuracy. The image quality scale showed more variability in individual image perception between spine surgeons and radiologists (ICC range 0.51–0.91). This study indicates that intraoperative screw positioning can be reliably assessed on CBCT for augmented reality surgical navigation when using optimized image quality. Subjective image quality was rated slightly superior for MAR compared to NoMAR.

## 1. Introduction

Over the course of the last two decades, the proportion of thoracolumbar Minimally Invasive Surgery (MIS) has increased for degenerative, traumatic, and metastatic indications of instrumented fusion techniques. The main benefits of MIS are reduction of blood loss, shorter hospital stay, improved short-term clinical outcomes, and cost-effectiveness [1,2]. Percutaneous pedicle screw misplacement represents one of the main complications in MIS procedures. Computer-assisted navigation with intraoperative three-dimensional (3D) imaging has been shown to be superior compared to conventional fluoroscopy-guided pedicle screw placement [2,3]. Augmented reality surgical navigation (ARSN) is a relatively recent technique, which utilizes video cameras in the tracking of non-invasive markers placed on the skin. Imaging of the spine used for ARSN is based on cone beam computed tomography (CBCT), which enhances imaging quality and lowers radiation exposure to the operating room staff compared to fluoroscopy [4]. Preclinical studies on cadaveric specimens have demonstrated the feasibility and safety of ARSN [5,6]. A clinical trial on MIS Transforaminal Lumbar Interbody Fusion (TLIF) using ARSN demonstrated a 94% accuracy of percutaneous pedicle screw placement [7].

Nevertheless, accurate intraoperative judgment can be challenging as metal artifacts, such as blooming around the screw shaft and streaks at the screw tip, may prevent the proper assessment of screw placement [8,9]. Metal artifacts originate from data inconsistencies caused by strong, energy-dependent attenuation [10,11]. Different metal artifact reduction (MAR) algorithms were developed to improve the intraoperative CBCT imaging quality based on the replacement of the projection data within the metal shadow by interpolation from surrounding detector pixel values [12,13,14]. This method can also be extended by known-component image reconstruction [13,14]. Another technique uses optimized C-arm orbits in order to avoid collinearity between the X-ray source and screws [15,16]. Figure 1 and Figure 2 demonstrate ARSN images for the baseline technology without MAR (NoMAR) in comparison to images using MAR, which is based on the interpolation of projection data neighboring the metal shadow around the pedicle screw [10]. The first reconstruction (NoMAR) is used to segment the metal voxels based on their brightness. The metal voxels are projected forward into the geometry of the measured projection, thus defining a 2D “metal shadow”. Information from pixels surrounding the metal shadow is used to interpolate “metal-free” projection data. A supplementary step is added to reinsert information of smaller overlaying structures in the interpolated data. A second reconstruction is performed from the “metal-free” projection data. Finally, a modified copy of the 3D titanium screw from the first to the second reconstructed image (MAR) aims to improve the appearance of the transition between the metal structure and the background.

We hypothesized that the intraoperative CBCT imaging gathered from MIS-TLIF procedures using ARSN would allow accurate and reproducible pedicle screw placement assessment and that the perception of image quality might improve between NoMAR and MAR.

The primary aim of this study was to assess intra- and inter-observer reliability of pedicle screw placement using intraoperative CBCT between spine surgeons and interventional radiologists with expertise in spinal procedures. The secondary aim was to compare the perception of image quality between NoMAR and MAR.

## 2. Materials and Methods

### 2.1. Patient Population

Institutional review board approval (CE-2020-90) was obtained for this retrospective study evaluating demographic and intraoperative radiologic data. Twenty-four consecutive patients with degenerative spondylolisthesis who underwent single-level decompression and MIS-TLIF using ARSN fusion at L3-L4 or L4-L5 between January and December 2019 were included. Patients with previous surgery (except microdiscectomy) and implants in the lumbar spine or in the immediate neighborhood, as well as patients requiring more than single-level MIS fusion, were excluded.

### 2.2. Surgical Procedure

The surgeries were performed in a hybrid operating room. The patient was positioned prone on a carbon table (Maquet, Getinge, Germany). Percutaneous pedicle screw placement was performed using ARSN (AlluraClarity Flexmove, Philips, The Netherlands) equipped with a robotic C-arm comprising a flat panel detector for 2D and 3D X-ray imaging, and a video camera system for tracking adhesive fiducials on the skin in the surgical field [5,6,7,8]. An initial X-ray identifying the lumbar region of interest was obtained, followed by iso-centering and a 10 s rotation CBCT image acquisition. The scans were performed with 20-degree tilted and non-tilted circular C-arm orbits. The acquisitions were performed with an X-ray dose modulation with exposure values varying between 1.8 mAs and 3.2 mAs per projection. Thus, the total CBCT scan dose with 300 projections was around 700 mAs. The vertebrae and corresponding pedicles were automatically segmented on the planning CBCT scan. The screw trajectories, diameters, and lengths were then intraoperatively planed on multiplanar CBCT views. The planned screw entry points and trajectories were then augmented to video images showing the surgical field for navigation of the Jamshidi needles. After placement of all the Jamshidi needles and K-wires, 6.5 × 45 mm diameter by length cannulated screws (ES2, Stryker, Allendale, NJ, USA) were manually placed. A control CBCT was then performed to check the screw placement.

### 2.3. Image Analysis

The CBCT screw control images were retrospectively extracted in Digital Imaging and Communications in Medicine (DICOM) format and anonymized for further evaluation. The images of 24 patients were treated using NoMAR and MAR by an independent engineer, thus creating 48 randomized files. The images were then independently analyzed by 3 spine surgeons and 3 radiologists, with expertise in percutaneous interventional spinal applications, using the RadiAnt DICOM Viewer 2021.2.2 software. Each observer analyzed the images on multiplanar reconstructions twice, at an interval of one month, in random order.

Screw positioning accuracy within the pedicle was rated using the Gertzbein and Robins classification [17]:*Grade 0*: screw within the pedicle without cortical breach;*Grade 1*: 0–2 mm breach, minor perforation, including cortical encroachment;*Grade 2*: 2–4 mm breach, moderate breach;*Grade 3*: >4 mm breach, severe displacement, which was not found in this cohort.

The screw diameter of 6.5 mm was used as a scale, which enabled the measurement of cortical breaches on axial and coronal views of each pedicle, as demonstrated in Figure 3.

The perception of image clarity around the screw contours was rated using an image quality scale based on the following criteria (Figure 4):*Grade 0*: mild artifact, screw thread clearly visible, cortical bone clearly visible, bone soft tissue interface clearly visible, clear interpretation possible;*Grade 1*: moderate artifact, screw thread visible, cortical bone with unclear portions, bone soft tissue interface distinguishable, interpretation possible;*Grade 2*: strong artifact, screw thread unclear, cortical bone contours unclear, bone soft tissue not distinguishable, interpretation difficult.

### 2.4. Statistical Analysis

Statistical analysis was performed using R Software 3.6.0 (R Foundation for Statistical Computing, Vienna, Austria). The reliability of rating screw positioning according to Gertzbein and Robins was compared between the observers, and for each observer, between both assessments. Likewise, the image quality scale was tested using Intraclass Correlation Coefficients (ICC) for categorical data. Observer agreement was then compared between NoMAR and MAR. The values of intraclass correlation coefficients were rated according to Landis and Koch [18]: correlation was considered excellent if r > 0.80, good if r = 0.61 to 0.80, fair if r = 0.41 to 0.60, and poor if r ≤ 0.40.

## 3. Results

### 3.1. Demographics

Twenty-four consecutive patients with degenerative spondylolisthesis were enrolled. L4-L5 was instrumented in 17 patients (70.8%) and L3-L4 in 6 patients (19.2%). There were 18 females and 6 males with an average age of 68.2 ± 9.1 years and Body Mass Index (BMI) of 25.9 ± 4.2 kg/m^2^. A total of 96 percutaneous pedicle screws, 4 per patient, were analyzed twice by 6 observers, which led to 1152 ratings for NoMAR and MAR imaging modalities, respectively.

### 3.2. Gertzbein and Robins Classification

The distribution of 1152 screw accuracy ratings comparing NoMAR versus MAR, respectively, was: 73.9% versus 69.9% for *Grade 0*, 20.2% versus 22.9% for *Grade 1*, and 5.9% versus 7.2% for *Grade 2.* Table 1 demonstrates ICC values for each observer individually and between the different observers for the Gertzbein and Robins classification. A good intra-observer agreement between the first and the second rating was demonstrated by each observer. An excellent overall inter-observer agreement was demonstrated by the six observers for the screw accuracy rating. Minor differences in ICC values existed between surgeons and radiologists. The ICC values were similar between NoMAR and MAR for screw accuracy ratings.

### 3.3. Image Quality Scale

The distribution of 1152 image quality ratings comparing NoMAR versus MAR, respectively, was: 21.2% versus 43.5% for *Grade 0*, 56.5% versus 41.7% for *Grade 1*, and 22.3% versus 14.8% for *Grade 2.*
Table 2 displays the ICC values for the image quality scale. Image quality perception was rated consistently between the first and the second assessment by all observers. Intra-observer agreement ranged from fair to excellent between the different observers. The ICC differences were small between NoMAR and MAR for each observer, respectively. Overall, image quality perception yielded good inter-observer agreement for NoMAR and MAR. Differences in ICC values existed between surgeons and radiologists.

## 4. Discussion

The assessment of pedicle screw accuracy represents a major concern for spine surgeons, as inaccurate placement may result in injury to various anatomical structures surrounding the pedicle, including the nerve roots, the spinal cord, and the blood vessels. Intraoperative assessment is suitable during MIS, as the procedure usually relies on fluoroscopy [19,20,21] or CBCT [6,7,8,9]. Hohenhaus et al. [3] and Lu et al. [4] compared screw accuracy between fluoroscopic and CBCT guidance and demonstrated the advantages of CBCT for precise screw positioning. Scarone et al. [22] and Hecht et al. [23] showed that intraoperative CT imaging enabled the accurate placement of pedicle screws in 95% of cases. Similar rates were reported for ARSN based on CBCT [6,7], which is associated with a lower radiation dose to the patient compared to CT [8]. However, the most common imaging study used for the exact assessment of the screw position within the pedicle is CT. Elliot et al. [24] demonstrated characteristics of titanium pedicle screw imaging using CT on 20 patients, with a total of 151 screws in clinical routine. All screws were measured larger than the known screw diameter. However, this discrepancy was less than 1 mm. Furthermore, the screw dimensions influenced the extent of the metal artifact, with different scatter amounts and more bloom artifact when screw size was increased. Moreover, the imaging technology itself and CT settings may influence the image quality around the pedicle screw. Huber et al. [11] compared different CT MAR strategies in an animal study on sheep using titanium pedicle screws. They demonstrated differences in MAR when comparing single versus dual-energy CT, different tube voltages at comparable radiation doses, and iterative image reconstruction versus monoenergetic extrapolation. CT is interesting when studying screw accuracy postoperatively. Modern intraoperative CBCT imaging, providing high image quality on bony structures and metal implants, was used in the current study. Normalized MAR techniques derived from CT [10] were applied to the CBCT technology, aiming for further improvement of intraoperative imaging quality.

Two main principles can be applied for improving intraoperative CBCT imaging of pedicle screws: optimized C-arm trajectories and MAR. Optimized C-arm orbits may be used to avoid collinearity between the X-ray beam and the direction of the screws to minimize metal artifacts in the 3D images. Klingler et al. [7] compared two mobile CBCT technologies using 135° elliptic scanning versus 190° isocentric scanning, which led to similar performances regarding the delineation of cortical bone in the thoracolumbar spine. Elliptic scanning elicited fewer metal artifacts compared to isocentric scanning. Wu et al. [15] compared MAR with different C-arm trajectories in reducing blooming artifacts on CBCT reconstructions and demonstrated that non-circular orbits reduced metal artifacts by 46% compared to circular orbits. Thies et al. [16] introduced a MAR technique that uses non-circular C-arm orbits with intraoperative adjustments of X-ray source trajectory to optimize the image reconstruction quality. These adjustments are based on a machine-learning algorithm using convolutional neural networks, which can predict quality metrics that enable scene-specific adjustments of the CBCT source trajectory, thus improving the image quality and reducing metal artifacts. In our study, circular C-arm orbits with a 20-degree tilt and without tilt have been used. In some patients, differences in the number of metal artifacts were observed, particularly between cranial and caudal pedicle screws, which is probably related to the differences between the orientation of the X-ray beam and the screw axis in the transverse plane. However, this factor was not clearly evaluated.

Software-based MAR algorithms may be used to mitigate streak and blooming artifacts, typically by iterative reconstructions. Privalov et al. [12] acquired data on a mobile CBCT system and compared different iterative MAR algorithms by assessing the visibility of pedicle walls and of the anterior and posterior vertebral body. On a scale from 0 to 4, a moderate improvement from 1.7 to 2.0 for the best MAR algorithm was achieved. Zhang et al. [13] demonstrated that a known-component 3D CBCT image reconstruction used for surgical navigation with the O-arm resulted in a 66.3% reduction in blooming artifacts around the screw shaft and provided a 65.8% decrease in streaks at the screw tip. This MAR allowed a clearer depiction of the screw within the pedicle and the vertebral body. Likewise, Uneri et al. [14] used a known-component MAR and demonstrated a reduction in pedicle screw and rod artifacts by 40% to 80% in a cadaver study. In our study, both NoMAR and MAR provided image reconstructions that allowed a reliable assessment of the screw position in most cases. The subjective image quality performances were rated superior for the MAR imaging modality. A clearcut depiction of metal and bony structures was possible with both algorithms at different quality levels. Nevertheless, the screw position accuracy was rated consistently with good to excellent ICC values by radiologists and spine surgeons using the Gertzbein and Robins criteria, which are based on 2 mm incremental steps of cortical encroachment or breaches by the screw [17].

The present pilot study has limitations as it compares the intraoperative CBCT data collected retrospectively during a limited consecutive series of 24 patients operated on using ARSN. The imaging data focused on the operated and adjacent lumbar segments. The surrounding abdominal organs, paravertebral muscles, subcutaneous fat thickness, and bone mineral density may play a role, and the relationship between the patient’s BMI and image quality could not be evaluated retrospectively. An additional validation study might integrate these factors, which could influence image quality, aiming for an optimal patient-specific setting for image acquisition and evaluation.

## 5. Conclusions

Intraoperative screw positioning can be reliably assessed on CBCT for ANSR when using optimized image quality, with good intra- and excellent inter-observer correlation for surgeons and radiologists. Subjective image quality was rated slightly superior for MAR compared to NoMAR.

## Figures and Tables

**Figure 1 sensors-22-04615-f001:**
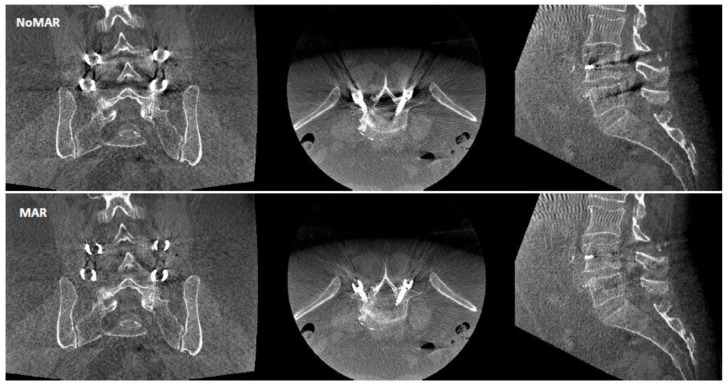
Image quality for one patient using CBCT reconstruction without metal artifact reduction algorithm (NoMAR) and with metal artifact reduction algorithm (MAR) based on interpolation of projection data surrounding the metal shadow around the pedicle screws.

**Figure 2 sensors-22-04615-f002:**
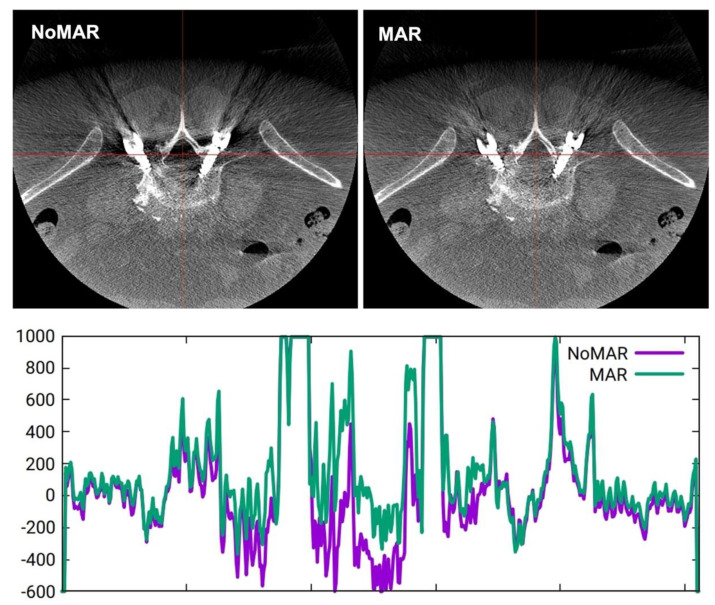
Variation of Hounsfield units across a coronal plane section (red line) comparing NoMAR (purple curve) and MAR (green curve).

**Figure 3 sensors-22-04615-f003:**
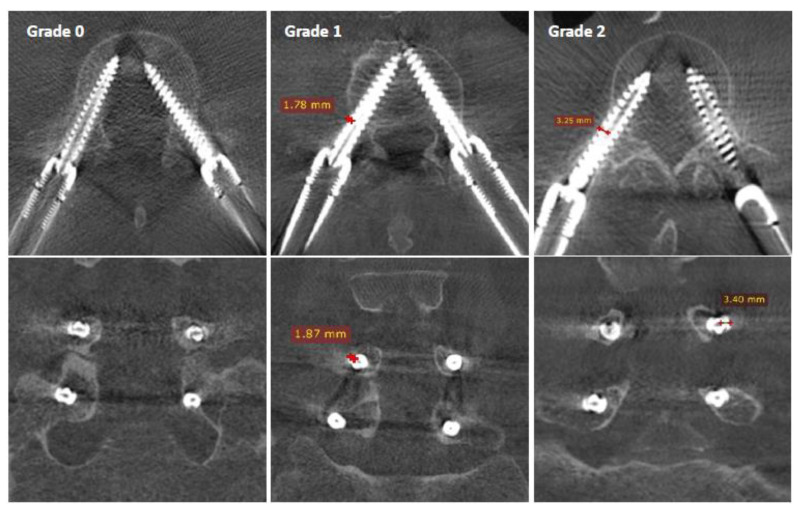
Screw placement accuracy with the pedicle according to Gertzbein and Robins: Grade 0 is perfectly contained within the pedicle, Grade 1 represents a cortical encroachment <2 mm, and Grade 2 is a breach between 2 and 4 mm.

**Figure 4 sensors-22-04615-f004:**
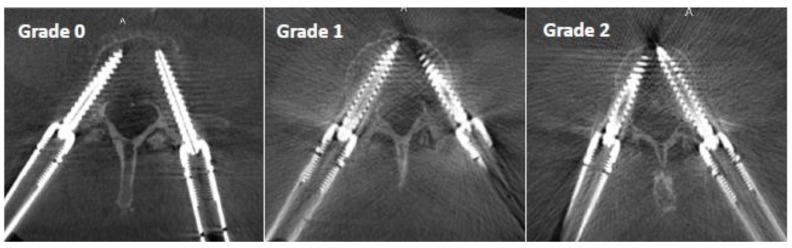
Pedicle screw metal artifact quality scale: Grade 0 represents a clearcut image of the screw and bony structures, Grade 1 allows assessment of the screw position with some limitations due to artifacts, and Grade 2 has a strong artifact which represents a limitation for the assessment of the exact screw position.

**Table 1 sensors-22-04615-t001:** Intra-class correlation coefficients and 95% credible intervals for the Gertzbein–Robins classification without (NoMAR) and with metal artifact reduction algorithm (MAR).

	NoMAR	MAR
** *Intra-Observer Correlation* **	** *ICC [95% CI]* **	** *ICC [95% CI]* **
Observer 1—surgeon	0.63 [0.45–0.77]	0.67 [0.50–0.79]
Observer 2—surgeon	0.84 [0.74–0.90]	0.78 [0.65–0.86]
Observer 3—surgeon	0.77 [0.63–0.86]	0.58 [0.37–0.73]
Observer 4—radiologist	0.90 [0.83–0.94]	0.85 [0.75–0.91]
Observer 5—radiologist	0.79 [0.67–0.87]	0.82 [0.71–0.89]
Observer 6—radiologist	0.85 [0.75–0.91]	0.90 [0.84–0.94]
** *Inter-observer Correlation* **		
Between 3 surgeons	0.89 [0.83–0.93]	0.91 [0.86–0.95]
Between 3 radiologists	0.74 [0.59–0.84]	0.61 [0.39–0.75]
Between all 6 observers	0.88 [0.82–0.92]	0.85 [0.78–0.90]

**Table 2 sensors-22-04615-t002:** Intra-class correlation coefficients and 95% credible intervals for the image quality scale without (NoMAR) and with metal artifact reduction algorithm (MAR).

	NoMAR	MAR
** *Intra-observer Correlation* **	** *ICC [95% CI]* **	** *ICC [95% CI]* **
Observer 1—surgeon	0.73 [0.58–0.83]	0.58 [0.38–0.73]
Observer 2—surgeon	0.79 [0.66–0.87]	0.69 [0.53–0.81]
Observer 3—surgeon	0.64 [0.38–0.85]	0.53 [0.23–0.75]
Observer 4—radiologist	0.95 [0.91–0.97]	0.87 [0.79–0.92]
Observer 5—radiologist	0.63 [0.43–0.76]	0.69 [0.53–0.81]
Observer 6—radiologist	0.51 [0.29–0.68]	0.58 [0.34–0.69]
** *Inter-observer Correlation* **		
Between 3 surgeons	0.81 [0.70–0.88]	0.70 [0.38–0.84]
Between 3 radiologists	0.62 [0.35–0.78]	0.54 [0.28–0.71]
Between all 6 observers	0.72 [0.55–0.80]	0.63 [0.48–0.79]

## Data Availability

Data can be available from the corresponding author on reasonable request.

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
