# Peer review of "Accuracy Assessment of Percutaneous Pedicle Screw Placement Using Cone Beam Computed Tomography with Metal Artifact Reduction"

_sensors, 2022, doi:10.3390/s22124615_

Round 1

Reviewer 1 Report

- This study, which is focused on to assess intra- and inter-observer reliability of pedicle 17 screw placement and to compare the perception of baseline image quality (NoMAR) with optimized 18 image quality (MAR) in cone-beam computed tomography which can be interesting for the reader of the journal.

-  The writing is sometimes difficult to understand due to poor grammar and improper choice of words; a thorough language review is needed before it can be considered for publication.

- The introduction indicates the problem with existing streak artifacts in screws especially in CBCT.  The authors used an original approach to address this issue by MAR and non-MAR. This method is well known for dental CBCT imaging. However, the readers may not familiar to these distributions, thus I suggest (even with an supll. File) the should be a  brief description of the extreme value distributions for the article.

- The authors stated that they used “the iodine solution simulating the density of dentin”, please explain why they did use iodine solution which is important for reproduction of the study.

-It is not clear how the authors performed the patient population. Did they use of a power analysis for finding the patient population? If so please add.

- One concern is the degree of the MAR. As these it is computer generated algorithms, there is levels of reduction in terms of technical aspect. Thus, these issues should be included to the article in detail.

- I would like to see a profile of CBCT values. Although the CBCT values fluctuate greatly around the streak artifacts, it should be difficult to identify the exact positions of the artifacts on each CBCT‐value profile because the CBCT‐value variations caused by the artifacts cannot be distinguished from the other nonspecific image noises. Thus, I would like see additional profile calculations/graph.

- One limitation of this paper is that as authors stated there is only one CBCT device. In previous research, it was known that for mA s values larger than 100 MAs, the location parameters were almost constant regardless of the kinds of reconstruction kernel, Hence, I highly doubt if the method can be applied for other CBCT units. This issue should be also discussed.

Author Response

- This study, which is focused on to assess intra- and inter-observer reliability of pedicle 17 screw placement and to compare the perception of baseline image quality (NoMAR) with optimized 18 image quality (MAR) in cone-beam computed tomography which can be interesting for the reader of the journal.

-  The writing is sometimes difficult to understand due to poor grammar and improper choice of words; a thorough language review is needed before it can be considered for publication.

            The second author is native speaker. She reviewed and edited the entire manuscript.

- The introduction indicates the problem with existing streak artifacts in screws especially in CBCT.  The authors used an original approach to address this issue by MAR and non-MAR. This method is well known for dental CBCT imaging. However, the readers may not familiar to these distributions, thus I suggest (even with an supll. File) the should be a  brief description of the extreme value distributions for the article.

A brief description of main artifacts around and at the tip of pedicle screws are mentioned in the introduction. Streak and blooming artifacts represent both types that are clinically relevant.

            “Accurate intraoperative judgment can be challenging as metal artifacts, such as blooming around the screw shaft and streaks as the screw tip, may prevent proper assessment of screw placement [8]. “

We feel it could be misleading to mention dental CBCT as the present study and the focus issue in Sensors deals with virtual reality imaging and spine surgery.

An additional figure 2 was added to improve the characteristics of MAR vs NoMAR.

- The authors stated that they used “the iodine solution simulating the density of dentin”, please explain why they did use iodine solution which is important for reproduction of the study.

            We couldn’t find the words “the iodine solution simulating the density of dentin” in our manuscript.

-It is not clear how the authors performed the patient population. Did they use of a power analysis for finding the patient population? If so please add.

            All consecutive patients who were operated using a single level TLIF-fusion with ASNR were included. A total of n=1152 ratings were performed. Since this retrospective data set was very large, no additional sample size calculation was made. The following details are mentioned at the beginning of the Results.

“A total of 96 percutaneous pedicle screws, 4 per patient, were analyzed twice by 6 observers, which led to 1152 ratings for NoMAR and MAR imaging modalities respectively.“

- One concern is the degree of the MAR. As these it is computer generated algorithms, there is levels of reduction in terms of technical aspect. Thus, these issues should be included to the article in detail.

            The algorithm does not contain a scalar parameter to weigh the contribution of a correction image. The 3D reconstruction is a non-local operation and hence it can be either turned on or off. This is different e.g. for noise reduction algorithms, which estimate the contribution of Poisson noise. The “correction image” then might be weighted only with e.g. 50%.

We elaborated on the description of the MAR algorithm itself (see Introduction / Methods).

The first reconstruction (NoMAR) is used to segment the metal voxels based on their brightness. The metal voxels are forward projected into the geometry of the measured projection, thus defining a 2D “metal shadow”. Information from pixels surrounding the metal shadow is used to interpolate “metal-free” projection data. A supplementary step is added to reinsert information of smaller overlaying structures in the interpolated data. A second reconstruction is performed from the “metal-free” projection data. Finally, a modified copying of the 3D titanium screw from the first to the second reconstructed image (MAR) aims to improve the appearance of the transition between the metal structure and the background.

- I would like to see a profile of CBCT values. Although the CBCT values fluctuate greatly around the streak artifacts, it should be difficult to identify the exact positions of the artifacts on each CBCT‐value profile because the CBCT‐value variations caused by the artifacts cannot be distinguished from the other nonspecific image noises. Thus, I would like see additional profile calculations/graph.

We have added a new Fig. 2 which demonstrates differences in a coronal plane comparing Hounsfield units between MAR and NoMAR.  

- One limitation of this paper is that as authors stated there is only one CBCT device. In previous research, it was known that for mA s values larger than 100 MAs, the location parameters were almost constant regardless of the kinds of reconstruction kernel, Hence, I highly doubt if the method can be applied for other CBCT units. This issue should be also discussed.

“The acquisitions were performed with an X-ray dose modulation with exposure values varying between 1.8 mAs to 3.2 mAs per projection. Thus, the total CBCT scan dose with 300 projections was around 700 mAs. “

However, in our study we didn’t modify the reconstruction kernel between the methods, which mainly affects noise appearance and sharpness. We agree, that the X-ray dose is an important parameter and included the description above in the materials and methods section of the manuscript. Though, we expect that the improvement in image quality that we observed can be reproduced on CBCT systems with similar dose settings.

Reviewer 2 Report

This manuscript reported a study on the reliability of multiple raters in the pedicle screw placement using CBCT images with metal artifact reduction. The topic is interesting. The method is sound. The conclusion was cautiously drawn. 

My concern are the followings: 

(1) Using G-R classification to rate the accuracy of the screw placement, both inter and intra rater reliability were evaluated and compared via ICC.  I am wondering how much it may tell from the variation of the values of ICC in each case, considering Lands and KOch's view and others?

(2) Although 1152 images were assessed, the CBCT scans involved only 24. Wondering how strong the conclusion can be made on the effect of the metal artifacts reduction. In this sense, this is a preliminary study.

(3) The conclusion is vague. It is better to directly answer the questions. Please refer back to the hypothesis, line 71-78. 

Author Response

This manuscript reported a study on the reliability of multiple raters in the pedicle screw placement using CBCT images with metal artifact reduction. The topic is interesting. The method is sound. The conclusion was cautiously drawn. 

My concern are the followings: 

(1) Using G-R classification to rate the accuracy of the screw placement, both inter and intra rater reliability were evaluated and compared via ICC.  I am wondering how much it may tell from the variation of the values of ICC in each case, considering Lands and KOch's view and others?

It might be interesting to compare pedicles one by one as an individual observation. However, ICC values reflect the curve slope across a cloud of points with ideal concordance if the value is 1 and no concordance if it is 0. Thus we need to stick to the entire data set if we want to use ICC values for statistical evaluation rather than individual description.

(2) Although 1152 images were assessed, the CBCT scans involved only 24. Wondering how strong the conclusion can be made on the effect of the metal artifacts reduction. In this sense, this is a preliminary study.

The number of patients is relatively small, but the number of assessments is high (n=96 pedicles). We have added the limitation to the end of the discussion and suggest validation on a larger cohort taking patient specific factors into account. Our study, might however be used as a pilot study.

(3) The conclusion is vague. It is better to directly answer the questions. Please refer back to the hypothesis, line 71-78. 

We modified the conclusion in the abstract and in the manuscript:

“Intraoperative screw positioning can be reliably assessed on CBCT for ANSR when using optimized image quality with good intra- and excellent inter-observer correlation for surgeons and radiologists. Subjective image quality was rated slightly superior for MAR compared to NoMAR.”

Round 2

Reviewer 1 Report

The relevnat changes were made. Thus can eb accepeted as an orginal contribution.